# Focal Adhesion Maturation Responsible for Behavioral Changes in Human Corneal Stromal Fibroblasts on Fibrillar Substrates

**DOI:** 10.3390/ijms25168601

**Published:** 2024-08-07

**Authors:** Kirill E. Zhurenkov, Arseniy A. Lobov, Natalya B. Bildyug, Elga I. Alexander-Sinclair, Diana M. Darvish, Ekaterina V. Lomert, Daria V. Kriger, Bozhana R. Zainullina, Alina S. Chabina, Julia I. Khorolskaya, Daria A. Perepletchikova, Miralda I. Blinova, Natalia A. Mikhailova

**Affiliations:** 1Institute of Cytology Russian Academy of Sciences, St. Petersburg 194064, Russia; kzhu576@aucklanduni.ac.nz (K.E.Z.); arseniylobov@gmail.com (A.A.L.); relapse@yandex.ru (N.B.B.); elga.aleks@gmail.com (E.I.A.-S.); darvishdi@mail.ru (D.M.D.); e.lomert@gmail.com (E.V.L.); daryamalikova@gmail.com (D.V.K.); chabinaalina123@gmail.com (A.S.C.); j.i.khorolskaya@gmail.com (J.I.K.); dasha_perepletch@mail.ru (D.A.P.); mira.blinova@mail.ru (M.I.B.); 2Department of Cytology and Histology, St. Petersburg State University, St. Petersburg 199032, Russia; 3Centre for Molecular and Cell Technologies, St. Petersburg State University, St. Petersburg 199032, Russia; zainullinazhana@gmail.com

**Keywords:** corneal stromal fibroblasts, fibrillar substrates, mechanotransduction, focal adhesions

## Abstract

The functioning of the human cornea heavily relies on the maintenance of its extracellular matrix (ECM) mechanical properties. Within this context, corneal stromal fibroblasts (CSFs) are essential, as they are responsible for remodeling the corneal ECM. In this study, we used a decellularized human amniotic membrane (dHAM) and a custom fibrillar collagen film (FCF) to explore the effects of fibrillar materials on human CSFs. Our findings indicate that substrates like FCF can enhance the early development of focal adhesions (FAs), leading to the activation and propagation of mechanotransduction signals. This is primarily achieved through FAK autophosphorylation and YAP1 nuclear translocation pathways. Remarkably, inhibiting FAK autophosphorylation negated the observed changes. Proteome analysis further confirmed the central role of FAs in mechanotransduction propagation in CSFs cultured on FCF. This analysis also highlighted complex signaling pathways, including chromatin epigenetic modifications, in response to fibrillar substrates. Overall, our research highlights the potential pathways through which CSFs undergo behavioral changes when exposed to fibrillar substrates, identifying FAs as essential mechanotransducers.

## 1. Introduction

The human cornea’s functioning largely depends on the maintenance of a healthy stratified epithelium and a corneal stroma structure. While the corneal epithelium mainly serves as an outermost protecting layer, the corneal stroma comprises approximately 80–85% of the entire corneal thickness, providing transparency, mechanical strength, and stability [1]. Corneal stroma transparency stems from its unique fibril structure and the organization of its extracellular matrix (ECM) [2,3]. Particularly, the corneal stromal ECM is represented by homogeneous collagen fibrils with a diameter ranging from 23 to 35 nm, which are finely packed and organized into perpendicular lamellae layers [4]. Up to 20% of the corneal stroma is populated by corneal stromal fibroblasts (CSFs), which are responsible for maintaining and remodeling the corneal stromal ECM [4]. Remarkably, these cells not only regulate the corneal stromal ECM structure and organization but also participate in the improvement of its transparency by synthesizing and depositing corneal crystallins (e.g., ALDH1A1, ALDH3A1) within their cytoplasm [5,6].

During the last few decades, it has been extensively demonstrated that various cells can sense and respond to multiple mechanical stimuli in their microenvironment [7,8,9,10]. Disorganization of the highly ordered corneal stromal ECM has been shown to significantly affect corneal stromal fibroblast (CSF) functioning, inducing the deposition of disordered collagen fibrils and reducing the rate of crystallin synthesis [11,12]. At the same time, CSF culturing on nano- or microgratings and nanofibrous electrospun scaffolds enables them to maintain cell functioning and results in the deposition of properly ordered ECM proteins [13,14,15,16]. These findings indicate the importance of mechanical cues in directing CSF behavior and functioning in the design of biomaterials used for corneal restoration. However, behavioral changes in CSFs on native fibrillar substrates are still poorly elucidated.

Here, we conducted a set of experiments to investigate the mechanisms behind behavioral changes in cultured human CSFs on native fibrillar substrates. Two fibrillar materials were studied, including decellularized human amniotic membrane (dHAM), which is extensively used in eye surgery [17,18], and designed fibrillar collagen film (FCF). After that, we thoroughly compared how these materials affect CSF fate through analysis of focal adhesion (FA) maturation, mechanotransduction initiation, and subsequent alterations in ECM remodeling-related gene expression, protein synthesis, and migration behavior. CSFs were found to form supermature FAs on FCF within the first 8 h of culturing, followed by earlier mechanotransduction activation and corresponding alterations in the Col1A expression, while dHAM did not promote similar behavior. Meanwhile, FAK phosphorylation inhibition using a selective FAK autophosphorylation inhibitor (pFAKi; FAK Inhibitor 14) negated the observed changes. Finally, we demonstrated the complex effect of native fibrillar substrates on the CSF proteomic profile using shotgun proteomics. Our study presents one of the possible pathways by which human CSFs could be influenced by fibrillar substrates and highlights the importance of a comprehensive scaffold selection for corneal restoration.

## 2. Results

### 2.1. Focal Adhesion Maturation Is Enhanced in Cultured Human CSFs on FCF

To understand the possible influence of fibrillar substrates, a homogenous population of human CSFs (see Appendix A for CSF flow cytometry profiles) was first cultured on three different substrates for 8 h to determine if fibrillar substrates trigger early FA maturation (Figure 1). Two substrates, dHAM and FCF, composed of fibrils with diameters of 224.2 ± 43.05 and 51.59 ± 17.61 nm, respectively, were compared to flat tissue culture polystyrene (TCPS) (Figure 1A,B). Meanwhile, the Young’s moduli of the chosen fibrillar materials accounted for 51.05 ± 17.48 and 39.51 ± 15.69 kPa for dHAM and FCF, respectively, closely resembling the mechanical properties of the human anterior corneal stroma (Appendix A) [19].

After the first 8 h of culturing, FAs were mainly represented in the CSF perinuclear space on TCPS and partly dHAM, while on FCF, they were predominantly spread throughout cells (Figure 1C). Furthermore, CSFs formed a significantly greater number of FAs (TCPS, 141.3 ± 39.18; dHAM, 232.9 ± 33.43; FCF, 286.5 ± 74.53; *p* = 0.0381) being cultured on FCF (Figure 1D). The area of these FAs (TCPS, 0.4 µm^2^ ± 0.05; dHAM, 0.46 µm^2^ ± 0.03; FCF, 0.5 µm^2^ ± 0.03; *p* = 0.0522) was also larger on FCF (Figure 1E). Being cultured on TCPS, CSFs mainly formed nascent adhesions, accounting for 61.52% ± 4.43, which can be seen in Figure 1F. Mature FAs on this substrate accounted for 34.93% ± 3.02, while supermature FAs were represented by a surprisingly large number, having 3.54% ± 1.43. On dHAM, FA distribution was similar to TCPS, with nascent adhesions, mature FAs, and supermature FAs accounting for 55.62% ± 2.52, 41.15% ± 2.52, and 3.23% ± 0.25, respectively. Remarkably, CSFs were found to form more mature FAs cultured on FCF. In this case, nascent adhesions and mature FAs accounted for 52.48% ± 1.87 and 43.47% ± 0.78, respectively. The number of supermature FAs was the most pronounced, with 4.05% ± 1.1.

The observed tendency was further maintained after 24 h of culturing, with CSFs culturing on FCF. At the same time, there was a substantial increase in the number of FAs in CSFs cultured on TCPS (Appendix A).

### 2.2. FCF Induces Changes in Cell Matrix Interaction-Related Protein Expression and Early Mechanotransduction in Cultured Human CSFs

To assess the influence of fibrillar substrates on the overall protein expression, we estimated the effect of FCF on a CSF proteomics profile using shotgun proteomics (Figure 2).

Proteome analysis revealed clear differences between CSFs cultured on FCF and TCPS (Figure 2A; see Dataset 1 for detail). FCF primarily promoted substantial changes in the expression of cell matrix interaction-related proteins, including integrin receptors (e.g., ITGAV, ITGB1, ITGB5) and FA (VCL) proteins (Figure 2B and Appendix A). The gene ontology (GO) cellular compartment enrichment analysis indicated FAs as the most involved compartment after CSFs were cultured on FCF (Figure 2C). Another group of significantly up-regulated proteins involves a vast number of chromatin organization (e.g., H1.2, H1.5, H2AC8, H2AC20) and RNA splicing-related (e.g., SRSF7, SRSF10, LSM7) proteins (Appendix A). Apart from that, FCF also resulted in protein expression changes in cytoskeleton reorganization-related proteins, such as CRIP1 and TMSB10 (Appendix A). Interestingly, in CSFs cultured on TCPS, the main up-regulated cellular pathways included proteins involved in glucose metabolism (HK1, HK2) and the citric acid cycle (DLST, SDHA) (Appendix A).

As soon as we observed significant changes in ECM-associated proteins via shotgun proteomics, we investigated possible alterations in epithelial-to-mesenchymal transition (EMT)-related and ECM remodeling-associated gene expression, which could appear in these substrates in a manner similar to that in nano- and microgrooved materials [14], as well as changes in corneal crystallin proteins, contributing to corneal transparency upon corneal ECM remodeling and wound healing (Figure 3) [20].

Figure 3A shows the distinct pattern observed for the ECM remodeling-related gene expression. Interestingly, *SNAI1* and *TWIST1* gene expression did not change in CSFs cultured on FCF during the whole period. Meanwhile, there was a 2-fold increase in the expression of *SLUG* and a 2.5-fold change in the expression of *ACTA2* after 4 days of culturing on the same substrate. dHAM demonstrated the opposite trend of decreasing the expression of *TWIST1* during the whole period and the expression of *ACTA2* after 8 days of culturing. There were no alterations in the expression of *SNAI1* and *SLUG* in CSFs cultured on dHAM over 8 days.

Connective tissue remodeling-associated gene expression also displayed a distinct tendency. Notably, there was a 6-fold increase in the expression of *CTGF* in CSFs cultured on FCF and a 4-fold increase in CSFs cultured on dHAM after the first 2 days of culturing (Figure 3B). This significant up-regulation after the first 2 days of culturing was already negligible after 4 days (Figure 3B). In contrast, *Col1A1* mRNA expression had a nearly 3-fold increase in CSFs cultured on FCF after 2 days of culturing, which remained almost the same throughout the period (Figure 3B). CSFs cultured on dHAM showed no significant changes in *Col1A1* mRNA expression (Figure 3B).

To identify whether fibrillar substrates can influence cell functions associated with corneal transparency, mRNA expression of *ALDH1A1* and *ALDH3A1* was also checked (Figure 3C) [6]. *ALDH1A1* mRNA expression decreased significantly in CSFs cultured on dHAM during the whole period, while a slight down-regulation was observed in CSFs cultured on FCF after 8 days of culturing (Figure 3C). *ALDH3A1* mRNA expression was down-regulated in CSFs on both substrates (Figure 3C).

Finally, to identify whether fibrillar substrates can induce early mechanotransduction, we compared the amount of pFAK and cytoplasmic/nuclear YAP1 in CSFs cultured on various scaffolds for 8 h via q-IFA (Figure 4).

On FCF, CSFs registered significant alterations in pFAK levels, with a nearly 2-fold increase to 177.6% ± 22.15 for TCPS (Figure 4A,C). Meanwhile, when they were cultured on dHAM, the amount of pFAK decreased to 59.13% ± 6.47 (Figure 4A,C). Similar tendencies were observed regarding changes in YAP1 expression (Figure 4B). Mainly, FCF promoted a 4.5-fold change in the nYAP1 expression of 458.98% ± 44.04 for TCPS and a 2.5-fold change in the cYAP1 expression of 279.61% ± 70.05 for TCPS (Figure 4D). dHAM displayed less pronounced changes, having a 1.5-fold change in the nYAP1 expression up to 160.33% ± 34.12 for TCPS, but demonstrated the most significant increase in the cYAP1 expression, seeing a nearly 4.5-fold change to 445.86% ± 28.66 for TCPS (Figure 4D). Correspondingly, CSFs cultured on FCF showed the highest nYAP1/cYAP1 ratio of 9:1, while CSFs cultured on dHAM displayed the lowest nYAP1/cYAP1 ratio of 1.8:1 (Figure 4E).

Unexpectedly, qRT-PCR did not reveal similar changes in mechanotransduction-associated gene expression compared to the results identified by q-IFA (Figure 4F), as these proteins are known to be regulated post-translationally [21]. CSFs cultured on FCF showed no changes in the expression of *FAK* and *YAP1* compared to TCPS. At the same time, CSFs cultured on dHAM displayed a pronounced decrease in the expression of *FAK* and *YAP1* during 8 days of culturing, which started after 4 days of being cultured on this substrate.

Such differences in mechanotransduction might affect cell motility. To address this, CSFs were cultured on dHAM and FCF for 24 h. After that, the migration patterns of the CSFs were mapped and compared to TCPS (Figure 5).

It should be noted that statistically significant differences were not observed for all tested conditions, and only tendencies can be drawn, as shown in Figure 5A. Particularly, CSFs cultured on dHAM moved slightly faster and traveled more considerable distances compared to TCPS and FCF, displaying velocities of 62.41 µm/h ± 16.61, 66.94 µm/h ± 6.97, and 57.61 µm/h ± 11.63 and distances traveled of 407.03 µm ± 78.65, 542.37 µm ± 72.97, and 444.43 µm ± 91.7 for TCPS, dHAM, and FCF, respectively. At the same time, E-max was higher with CSFs cultured on FCF (1.79 ± 0.48) compared to TCPS (1.29 ± 0.03) and dHAM (1.16 ± 34). The observed changes in CSF behavior were more pronounced when evaluating mean square displacement (MSD) (Figure 5B). The MSD for cells cultured on TCPS, dHAM, and FCF increased over time, suggesting that cells migrated more as time progressed, with dHAM showing the highest MSD, implying that it could be the most permissive substrate for cell migration (Figure 5B).

Summarizing these data, we can conclude that FCF can enhance early FA development, leading to the activation and propagation of mechanotransduction signals. Based on our data, we assumed that FAK autophosphorylation is critical for further mechanotransduction activation and propagation. To prove this, we performed experiments with pFAK targeting inhibition (pFAKi).

### 2.3. pFAK Inhibition Overrules Early Mechanotransduction in Human CSFs Cultured on FCF

To investigate the role of FA maturation in behavioral changes in CSFs during the first 8 h of culturing, FAK Inhibitor 14 (pFAKi; 1 µM), a small molecule inhibitor of Y397-FAK autophosphorylation, was utilized (Figure 6).

Western blot analysis of CSFs with 3 days of pFAKi treatment revealed a pronounced decrease in both FAK and pFAK protein levels (Figure 6A,B). Interestingly, pFAKi led to a substantial increase in cell area in CSFs cultured on TCPS and FCF compared to the control (Figure 6C). It also resulted in an increase in the nuclei area in CSFs cultured on all substrates (Figure 6D). Regarding FAs, pFAKi resulted in a pronounced decrease up to the level of TCPS in the FA number per cell (TCPS, 220.1 ± 44.12; dHAM, 153.3 ± 62.75; FCF, 274.6 ± 54.15; *p* = 0.0869) and FA area (TCPS, 0.52 µm^2^ ± 0.05; dHAM, 0.39 µm^2^ ± 0.06; FCF, 0.42 µm^2^ ± 0.03; *p* = 0.0405) in CSFs cultured on dHAM and FCF (Figure 6E–G). More significantly, pFAKi resulted in a decrease in the number of mature and supermature FAs compared to TCPS (Nascent adhesions, 53.32% ± 4.35; Mature FAs, 41.66% ± 3.51; Supermature FAs; 5.02% ± 1.01; *p* < 0.05) in CSFs cultured on dHAM (Nascent adhesions, 61.72% ± 6.36; Mature FAs, 36.12% ± 5.49; Supermature FAs; 2.16% ± 0.88; *p* < 0.05) and FCF (Nascent adhesions, 58.13% ± 2.04; Mature FAs, 38.96% ± 1.24; Supermature FAs; 2.91% ± 0.95; *p* < 0.05) (Figure 6H). Overall, pFAKi led to significant cell spreading and the formation of even stress fibers and large peripheral FA patches in CSFs cultured on all substrates (Figure 6E).

We then evaluated the influence of pFAKi on early mechanotransduction initiation in CSFs cultured on fibrillar substrates during the first 8 h (Figure 7).

Indeed, pFAKi overruled the influence of fibrillar substrates on early mechanotransduction activation in CSFs discussed above (Figure 4). pFAK (TCPS, 100% ± 21.72; FCF, 121.6% ± 31.47; *p* = 0.7574) and nYAP1 (TCPS, 100% ± 20.32; FCF, 152.18% ± 43.29; *p* = 0.2815) levels in CSFs cultured on FCF did not significantly diverge from TCPS values (Figure 7A–D). Interestingly, the amount of pFAK (TCPS, 100% ± 21.72; dHAM, 158.9% ± 50.12; *p* = 0.1977) was higher in CSFs cultured on dHAM compared to TCPS, while the nYAP1 (TCPS, 100% ± 20.32; dHAM, 108.59% ± 33.64; *p* = 0.9624) level did not diverge from TCPS values (Figure 7A–D). It should be highlighted that on both fibrillar substrates, the nYAP1/cYAP1 ratio decreased significantly, being 2.2:1 and 3.6:1 for dHAM and FCF, respectively (Figure 7E).

In addition, we found that pFAKi influences *CTGF* and *Col1A1* mRNA expression after 2 days of culturing, as these two genes had the most pronounced up-regulation in untreated cells. We observed downregulation of *CTGF* and *Col1A1* following pFAKi treatment, although it did not reach statistical significance (Appendix A). Despite this, there may be biological significance due to the consistent downregulation observed across experimental replicates.

Finally, we observed some effects of pFAKi on CSFs’ motility (Appendix A). CSFs started moving faster and farther on all substrates (mainly on dHAM), with velocities (*p* = 0.4184) of 69.67 µm/h ± 12.08, 76.08 µm/h ± 17.14, and 64.28 µm/h ± 13.13 and distances traveled (*p* = 0.1670) of 431.77 µm ± 71.25, 570.63 µm ± 98.32, and 541.23 µm ± 215.88 for TCPS, dHAM, and FCF, respectively. E-max values also increased on all substrates, with dHAM and FCF having higher values (TCPS, 1.48 ± 0.17; dHAM, 1.62 ± 0.31; FCF, 2.07 ± 0.71; *p* = 0.0.0603). When treated with pFAKi, cells on TCPS, dHAM, and FCF also exhibited an increase in MSD over time, indicating active migration. pFAKi treatment enhanced cell migration on dHAM and FCF substrates for around 15 h before a potential inhibition or saturation of migration could be found.

## 3. Discussion

Substrate mechanics have been widely shown to affect various aspects of cell fate and behavior [22,23]. The human cornea represents a highly complex tissue, acting as the first refractive lens of the eye [2]. Furthermore, multiple corneal functions rely on its mechanical properties [24]. For instance, Gouveia et al. demonstrated that matrix stiffening following alkali damage induced YAP1-dependent mechanotransduction, promoting Wnt/β-catenin signaling and suppressing Sox9 signaling, which resulted in limbal epithelial stem cell loss. In contrast, collagenase softening post-burn inactivated YAP1-dependent mechanotransduction, leading to a limbal niche recovery [25].

Corneal stroma comprises almost 80% of the corneal thickness and is composed of highly ordered nanofibrils [4]. Nanotopography has been demonstrated to affect the functions of CSFs, which are responsible for maintaining and remodeling the corneal stromal ECM [26]. However, the diverse effects on CSF fate and behavior still need to be better elucidated, especially regarding native fibrillar substrates.

In this study, we first demonstrated how human CSFs respond to fibrillar substrates like dHAM and custom FCF. We focused on cellular characteristics, beginning with the earliest cell response. Following this, we observed and reported specific alterations in cell behavior. We demonstrated that CSFs could form mature FAs already after 8 h of culturing on FCF, leading to significantly faster FAK autophosphorylation and YAP1 nuclear translocation.

Here, we consider FAK autophosphorylation one of the most critical steps, contributing to mechanotransduction propagation in response to fibrillar substrates. Primarily, when we inhibited its function using a selective Y397-FAK autophosphorylation inhibitor (FAK Inhibitor 14), it led to the complete disruption of early FA maturation on all fibrillar substrates, and the positive impact of FCF on FAK autophosphorylation and YAP1 nuclear translocation was overruled. Teo et al. observed a similar influence of pFAKi on mechanotransduction in human mesenchymal stem cells (hMSCs) cultured on nanogratings. In their study, 250 nm nanogratings were found to promote neurogenic and myogenic differentiation independently of any ECM proteins. The authors also demonstrated that the differentiation of hMSCs induced by topography is governed by a mechanism reliant on direct force, utilizing pFAK to convey topographical cues to the nucleus. They further revealed that this process is disrupted when either a Y397-FAK-specific inhibitor (PF573228) or FAK siRNA is used, effectively abrogating the impact of nanotopography on cultured hMSCs [27]. Meanwhile, Lachowski et al. verified that Y397-FAK autophosphorylation controls YAP1 nuclear translocation and activation in response to external mechanical stimuli in cultured primary human hepatic stellate cells [28].

Interestingly, we did not observe significant changes in FAK and YAP1 mRNA expression in CSFs cultured on FCF after 2, 4, or 8 days of culturing. This may indicate that mRNA expression can appear much earlier following FA maturation. For instance, Zarka et al. indicated the early response of MLO-Y4 cells to cyclic stretching within 9 h, which was confirmed by an increase in *YAP/TAZ* gene expression [29]. In contrast, *FAK* and *YAP1* mRNA expression in CSFs cultured on dHAM decreased over time. This suggests a lack of positive feedback in activating the mechanotransduction pathways in these cells. This reduction in gene expression could be attributed to the differences between the fibrillar topography of dHAM and the native fibrils found in the corneal stroma, implying that the physical environment substantially influences cellular responses, which also aligns with findings considering corneal cell culturing on nanogratings differing in ridge width [15,30,31].

Considering the more diverse impact of fibrillar substrates on cultured cells, some exciting findings were drawn in our research. Notably, there was pronounced up-regulation in *CTGF* and *Col1A1* mRNA expression after 2 days of culturing, as well as *SLUG* and *ACTA2* mRNA expression after 4 days of culturing in CSFs cultured on FCF. At the same time, *TWIST1* and *ACTA2* witnessed pronounced down-regulation in CSFs cultured on dHAM starting from the second day of culturing. Here, the observed changes could suggest a positive feedback mechanism. FCF appears to display signals that activate pathways involved in tissue remodeling and collagen production. In contrast, when CSFs are cultured on dHAM, there is a lack of gene expression changes typically associated with a reparative response. This indicates that dHAM may not provide the necessary mechanical cues to stimulate CSFs. The aforementioned genes are usually involved in EMT behavior, frequently followed by wound healing and fibrosis processes [32]. At the same time, *ACTA2* (α-smooth muscle actin, α-SMA) was shown to perform multiple functions in mechanotransduction, including a mechanotransducer role. For instance, α-SMA typically contributes to stress fibers in cells but can also enrich FAs, promoting their maturation, thereby linking the ECM and actin cytoskeleton [33]. Wan et al. demonstrated co-dependency between YAP/TAZ-associated mechanotransduction and the up-regulation of *CTGF* mRNA expression in human adipose-derived mesenchymal stem cells (ASCs) cultured on aligned poly (L-lactic acid) electrospun fibrous scaffolds [34]. Remarkably, *CTGF* and *Col1A1* gene expression alterations on fibrillar substrates were overruled by pFAKi, indicating the crucial role of FAs following mechanotransduction propagation in response to fibrillar topography.

Fibrillar substrates could also affect CSF migration behavior. We could not obtain statistical significance for the observed results, indicating the complicated nature of cell motility data analysis. However, some tendencies can be drawn. For example, when cultured on FCF, CSFs demonstrated the highest degree of movement (E-max) compared to other substrates. Meanwhile, cells displayed the longest distance travelled when cultured on dHAM. pFAKi resulted in an increase in E-max in CSFs cultured on both dHAM and FCF. The cell motility pattern analysis revealed more appealing details. Namely, it indicated that CSF movement on dHAM was more chaotic and largely displaced than on other substrates.

Finally, the larger-scale evaluation of fibrillar substrate’s influence on cultured CSFs was conducted via proteome analysis. Proteomics identified a few major protein clusters involved in cellular responses to FCF. Notably, it revealed FAs as the most affected cellular compartment, suggesting that FAs and FAK autophosphorylation are critical for mechanotransduction activation and propagation. Another interesting finding from the proteome analysis indicates the involvement of multiple proteins contributing to alternative splicing and chromatin epigenetic modification upon mechanical stimuli. For instance, Nemec and Kilian identified substrate mechanics critical for material control over cell epigenetics, including those imposed by non-coding RNAs, DNA CpG methylation, and histone acetylation and methylation [35]. More recently, Murugan et al. highlighted the importance of biophysical control of plasticity and patterning in regeneration and cancer [36].

Overall, our study underscores the significance of the scaffold choice in directing CSF fate and behavior. FAs emerge as crucial for the early activation and propagation of mechanotransduction pathways in CSFs on fibrillar substrates. Meanwhile, inhibiting one of the principal FA enzymes, such as FAK, results in mechanotransduction inactivation. We also demonstrate the diverse effects of specific fibrillar substrates, which resemble the native corneal stromal ECM, on various cellular behaviors, including gene expression and protein synthesis.

## 4. Materials and Methods

### 4.1. Human Tissue Sourcing

Human cadaveric donor corneas (*n* = 3) were obtained from S. M. Kirov Military Medical Academy (St. Petersburg, Russia) under the ethics approval of the Ethics Committee of the S. M. Kirov Military Medical Academy (approval no. 212). Amniotic membranes were obtained during planned Caesarean deliveries from healthy patients (*n* = 4). All procedures were performed following informed consent for use for research purposes.

### 4.2. Isolation and Culture of Human Corneal Stromal Fibroblasts

Immediately after the collection, human corneal stromal fibroblasts (CSFs) were isolated from the corneal stroma according to the following protocol (see Appendix A for detail). Briefly, collected corneas were washed three times with phosphate-buffered saline (PBS, pH 7.4, Ca^2+^ and Mg^2+^ free) containing antibiotics (250 µg/mL gentamicin; 1000 U/mL Pen/Strep; Life Technologies, Carlsbad, CA, USA). The remaining scleral and limbal tissues were dissected from the cornea, while the corneal epithelium and endothelium were scraped off using a surgical blade. After that, the corneal stroma was cut into small pieces and incubated in an enzyme solution (12 mg/mL Collagenase type I; Sigma-Aldrich, St. Louis, MO, USA) at 4 °C overnight. The next day, stromal samples were dissociated via vortexing for 20–30 s, centrifuged for 3 min at 1000× *g*, and the final cell suspension was plated in a culture dish. CSFs were cultured in DMEM/F-12 medium (Life Technologies) supplemented with 10% fetal bovine serum (FBS; Cytiva, Marlborough, MA, USA), 1000 U/mL Pen/Strep (Life Technologies), and 0.5 ng/mL amphotericin B (Thermo Scientific, Waltham, MA, USA) in an incubator at 37 °C, 5% CO_2_, and 95% humidity. One to two weeks after isolation, CSFs were tested for mycoplasma contamination. After reaching 70–80% confluency, cells were passaged using 0.25% trypsin-EDTA (Life Technologies) for 3 min at 37 °C and cultured for up to eight passages for further in vitro studies.

### 4.3. Flow Cytometry

The homogeneity of the derived cell culture was checked with flow cytometry using specific antibodies conjugated with phycoerythrin (PE) in the dilutions recommended by the manufacturer (Appendix A). Three independent experiments were performed on CSFs derived from three different donors. Before examination, CSFs (1 × 10^6^) were resuspended in 1 mL of PBS (pH 7.4, Ca^2+^ and Mg^2+^ free) and separately stained with appropriate primary conjugated antibodies in the dark at room temperature for 60 min. The stained cells were diluted with PBS (1:10) and examined using a CytoFLEX Flow Cytometer (Beckman Coulter, Brea, CA, USA). The isotype of PE (Iso PE) was used as a negative control.

### 4.4. Human Amniotic Membrane Preparation

Decellularized human amniotic membranes (dHAMs) were prepared from human placentas obtained during planned Caesarean deliveries and processed as previously reported [37,38]. Briefly, amniotic membranes were mechanically separated from the chorion, washed in Ringer’s solution containing ceftriaxone (10 mg/mL; Sintez, Moscow, Russia), fixed onto 3 cm Ø Petri dishes without bottoms under sterile conditions, and cryopreserved at −80 °C in a mixture of DMEM/F12 medium and dimethyl sulfoxide (1:1) (DMSO; BioloT, St. Petersburg, Russia). Before further use, dHAMs were thawed at 37 °C, washed three times with PBS (pH 7.4, Ca^2+^ and Mg^2+^ free), decellularized at 37 °C for 45 min using 0.25% Trypsin-EDTA (BioloT), and sterilized under UV-light irradiation for 3 h. For immunofluorescence analysis, dHAM was attached to round glass coverslips, dried for 2–3 days at 30 °C, and sterilized under UV-light irradiation for 3 h.

### 4.5. Fibrillar Collagen Film Fabrication

According to the previously published procedure, collagen type I was extracted from rat tail tendons using acid solubilization [39]. Briefly, rat tail tendons were solubilized in 0.5 M acetic acid and precipitated twice, first with 0.9 M NaCl and then with 0.02 M Na_2_HPO_4_. The precipitate was again solubilized in 0.5 M acetic acid with a subsequent reduction in the acid concentration to 0.1%. Finally, a 5 mg/mL concentration of collagen solution was obtained.

Fibrillar collagen films (FCFs) were prepared by mixing acidic collagen solution with 10× neutralizing salt solution containing 1.35 M NaCl, 0.3 M Na_2_HPO_4_, and 0.3 M (4-(2hydroxyethyl)-1-piperazineethanesulfonic acid) (HEPES) buffer to the final concentration of 0.46 mg/mL. The resulting solution was adjusted to pH = 7.4 with 1 M NaOH, drop-cast on round glass coverslips or 3/5 cm Ø Petri dishes, and dried at 30 °C for 24 h, as reported previously [40]. Prepared collagen films were rinsed three times with type 1 water (18.2 MΩ·cm, Milli-Q Direct 8, Type 1/ultrapure water), dried, and sterilized under UV-light irradiation for 3 h.

### 4.6. Scanning Electron Microscopy

Surface features of dHAM and FCF were imaged via scanning electron microscopy (SEM) with an FEI Phillips XL30 S-FEG instrument (Philips, Amsterdam, The Netherlands) after drying at 30 °C for 2–3 days and sputter-coating with a 5 nm layer of gold. Tissue culture polystyrene (TCPS) was imaged under the same conditions as a control. Fibril characterization was performed using ImageJ software (version 1.54f). Three samples were analyzed per condition, and at least 50 fibrils were quantified.

### 4.7. Immunostaining

CSFs (5 × 10^3^ cm^−2^) were seeded on prepared glass coverslips and cultured in 24-well plates in the supplemented DMEM/F-12 medium for 8 h. Cells then were washed three times with PBS (pH 7.4, Ca^2+^ and Mg^2+^ free), fixed with 4% paraformaldehyde (PFA; Sigma-Aldrich) for 20 min, rewashed with PBS, permeabilized with 0.1% Triton X-100 (Sigma-Aldrich) for 15 min, blocked in PBS supplemented with 1% bovine serum albumin (BSA; Thermo Scientific) at 37 °C for 1 h, and incubated overnight at 4 °C with primary antibodies (Appendix A) diluted in 0.1% PBS-Tween20 (PBST; Thermo Scientific) according to the manufacturer’s recommendations. Cells were then washed three times with PBST and incubated for 2 h at room temperature with the corresponding secondary antibodies in PBST. After that, samples were washed with PBST, stained with Phalloidin-TRITC (1:400; Thermo Scientific) for 25 min, rewashed with PBST, stained with DAPI (1 μg/mL; Thermo Fisher Scientific) for 2–3 min, washed three times with PBST, and mounted onto glass slides using ProLong™ Gold Antifade Mountant medium (Thermo Scientific). Images were collected using an OLYMPUS FV3000 confocal microscope (Olympus, Tokyo, Japan) or a CellVoyager™ CQ1 Benchtop High-Content Analysis System (Yokogawa Electric, Tokyo, Japan).

### 4.8. Quantitative Immunofluorescence Analysis

FA quantity, area, and circumference; cell cytoplasm and nuclear areas; and the intensity of the signal corresponding to cellular pFAK and YAP1 were imaged and analyzed using a CellVoyager™ CQ1 Benchtop High-Content Analysis System. For all data acquisition, imaging parameters were kept constant. Briefly, fluorophores were visualized via 405 nm (20%, 16 bit gain “low noise and high well capacity”, exposure 500 ms; excitation filter: 477/60 nm), 488 nm (20%, 16 bit gain “low noise and high well capacity”, exposure 500 ms; excitation filter: BP525/50 nm), and 561 nm (20%, 16 bit gain “low noise and high well capacity”, exposure 500 ms; excitation filter: BP617/73 nm) lasers in separate channels using a ×40 dry objective (NA = 0.95). Several *z*-sections (3 × 2.5-μm) were collected to obtain maximum signal intensities, while background noise was extracted automatically. The initial image type was black and white with a 2528 × 2136 pixel resolution, with one pixel equaling 0.1587 µm. To quantify FA, CSFs on the various substrates were immunofluorescently stained for the FA protein component paxillin, as described above (see Appendix A for analysis detail). The paxillin signal corresponding to the intracellular protein pool was extracted from the overall data using a circumference filter of more than 0.2 µm. After that, FAs were divided into several major fractions based on their circumference as described previously: nascent (0–2 µm), mature (2–6 µm), and supermature (>6 µm) [41]. To quantify YAP1 levels in the nucleus and cytoplasm, the corresponding subregions were identified. After that, the fluorescence levels of the nucleus and cytoplasm were subtracted from each other (see Appendix A for analysis detail). More than 1000 cells were analyzed per every condition, while at least three independent experiments were performed. Glass coverslips were used as a control.

### 4.9. Quantitative Real-Time Polymerase Chain Reaction

CSFs (1 × 10^4^ cm^−2^) were seeded on the prepared scaffolds and cultured in the supplemented DMEM/F-12 medium for 2, 4, and 8 days. The ExtractRNA reagent (Evrogen, Moscow, Russia) was used according to the manufacturer’s protocol to isolate RNA. The quality of the isolated RNA was verified by using 1% agarose gel electrophoresis to assess the integrity of the total RNA, and a Nanodrop 1000 spectrophotometer (Thermo Scientific) was utilized to ensure the 260/280 ratio was within the 1.8–2.0 range. Synthesis of cDNA from isolated total RNA (1 µg) was performed according to the manufacturer’s protocol using the MMLV RT kit (Evrogen) with random (dN)_10_ primers in a T100 Thermal Cycler (Bio-Rad Laboratories, Hercules, CA, USA). The quantitative polymerase chain reactions (qRT-PCR) were carried out according to the manufacturer’s protocol using qPCRmix-HS SYBR + LowROX (Evrogen) in a LightCycler^®^ 96 (Roche, Basel, Switzerland), with the following 40× three-step cycle: 10 s at 95 °C, 30 s at 60 °C, and 15 s at 72 °C. The transcription levels of *ACTA2*, *ALDH1A1*, *ALDH3A1*, *CCN2* (*CTGF*), *Col1A*, *PTK2* (*FAK*), *SLUG*, *SNAI1*, *TWIST1*, and *YAP1* using specific primers (Appendix A) were calculated by the delta–delta Ct (∆∆Ct) method and normalized to the expression of the *GAPDH* housekeeping gene. Data were represented as the gene expression relative to control cells (TCPS) from at least three independent experiments.

### 4.10. Cell Motility Analysis

CSFs (5 × 10^3^ cm^−2^) were seeded on the prepared scaffolds and cultured in the supplemented DMEM/F-12 medium for 24 h. Then, 2.5% Hoechst 33342 (Thermo Scientific) was added to cells for 10 min the next day. Fresh culture medium was then added to cells following several washings with PBS (pH 7.4, Ca^2+^ and Mg^2+^ free). Cell motility characteristics were analyzed via time-lapse imaging using a CellVoyager™ CQ1 Benchtop High-Content Analysis System, as previously reported [42]. Briefly, images were taken every 15 min within 24 h with a 405 nm laser and phase contrast microscopy using a ×10 dry objective (NA = 0.4). The initial image type was black and white with a resolution of 1264 × 1068 pixels, and one pixel equaled 1.3333 µm. Collected image sequences were then processed using the manual tracking plugin in ImageJ version 1.54f software, resulting in the subsets of *x*–*y* coordinates. At least 150 cells were manually tracked for every condition within three independent experiments (*n* = 3). Cell nuclei were used as cell centers. Obtained *x*–*y* coordinates were used to calculate the cell velocity, distance traveled, degree of cell movement (E-max), and mean square displacement (MSD), as well as to plot cell movement trajectories in the R environment following the previously described procedures [42,43]. TCPS was used as a control.

### 4.11. Proteomic Profiling

Shotgun proteomics analysis was performed to analyze the impact of FCF on CSF protein expression compared to TCPS after 8 days of culturing on a particular substrate. CSFs cultured on dHAM were not examined due to the higher probe contamination with matrix proteins. Shotgun proteomics analysis was carried out on the equipment of the Core Facility “Centre for Molecular and Cell Technologies” (St. Petersburg State University, St. Petersburg, Russia).

For the proteomic profiling, proteins were isolated from the same samples used for RNA analysis by acetone precipitation (Sigma-Aldrich). The protein pellet was resuspended in 8 mol/L urea (Sigma-Aldrich) diluted in 50 mmol/L ammonium bicarbonate (Sigma-Aldrich). The protein concentration was measured by a Qubit 4 fluorometer (Thermo Scientific) with the QuDye Protein Quantification Kit (Lumiprobe, Cockeysville, MD, USA) according to the manufacturer’s protocol. The protein quantification was verified by SDS-PAGE.

Protein samples (20 µg) were then incubated in 5 mmol/L dithiothreitol (Sigma-Aldrich) for 1 h at 37 °C with subsequent incubation in 15 mmol/L iodoacetamide (Sigma-Aldrich) for 30 min at room temperature. Then, the samples were diluted with 7 volumes of 50 mmol/L ammonium bicarbonate and incubated for 16 h at 37 °C with 400 ng of trypsin (Promega, Madison, WI, USA). The quality of protein digestion was verified by SDS-PAGE. Then, the sample was mixed with formic acid (Sigma Aldrich) to a 1% final concentration, evaporated in a Labconco Centrivap Centrifugal Concentrator (Eppendorf, Hamburg, Germany), and desalted with a C18 ZipTip (MilliporeSigma, Burlington, MA, USA) according to the manufacturer’s recommendations. Desalted peptides were evaporated and dissolved in water/0.1% formic acid for further LC-MS/MS analysis. We used two biological replicates for proteomics analysis. Each biological replicate was analyzed in technical triplicates.

Approximately 500 ng of peptides was used for shotgun proteomics analysis by UHPLC-MS/MS with ion mobility in a TimsToF Pro mass spectrometer (Bruker Daltonics, Bremen, Germany) with a nanoElute UHPLC system (Bruker Daltonics). UHPLC was performed in two-column separation mode with an Acclaim™ PepMap™ 5 mm Trap Cartridge (Thermo Scientific) and a Bruker Ten separation column (C18 ReproSil AQ, 100 mm × 0.75 mm, 1.9 µm, 120 Å; Bruker Daltonics) in gradient mode with a 500 nL/minute flow rate at 50 °C. Phase A was water/0.1% formic acid; phase B was acetonitrile/0.1% formic acid. The gradient was from 2% to 35% phase B for 38 min, with a subsequent wash with 95% phase B. Before each sample, the columns were equilibrated with ten and four column volumes. The CaptiveSpray ion source was used for electrospray ionization with 1600 V of capillary voltage, 3 L/min N_2_ flow, and 180 °C source temperature. The mass spectrometry acquisition was performed in automatic DDA PASEF mode with a 0.5 s cycle in positive polarity with the fragmentation of ions with at least two charges in the *m/z* range from 100 to 1700 and an ion mobility range from 0.85 to 1.30 1/K0.

Protein identification was performed in PEAKS Studio Xpro software (https://www.bioinfor.com/peaks-software/; a license granted to St. Petersburg State University; Bioinformatics Solutions Inc., Waterloo, ON, Canada) using the human protein SwissProt database (https://www.uniprot.org/; accessed on 20 July 2022; organism: Human [9606]; uploaded on 2 March 2021; 20,394 sequences) and protein contaminant database CRAP (version of 4 March 2019). The search parameters were: parent mass error tolerance of 10 ppm and fragment mass error tolerance of 0.05 ppm, protein and peptide FDR < 1% and 0.1%, respectively, and two possible missed cleavage sites. Cysteine carbamidomethylation was set as a fixed modification. Methionine oxidation, N-terminal acetylation, and asparagine and glutamine deamidation were set as variable modifications.

The mass spectrometry proteomics data were deposited to the ProteomeXchange Consortium via the PRIDE [44] partner repository with the dataset identifiers PXD048096 and 10.6019/PXD048096. Label-free quantification by peak area under the curve was used for further analysis in R (version 4.2.3; R Core Team, 2019) [45]. First, the filtration, Log-transformation, median normalization, and missed values imputation were performed with the NAguideR package [46]. Proteins with more than 50% of missed values in any biological group or coefficient of variation higher than 0.7 were removed. Then, the optimal method for imputation was chosen based on the “classic” criteria, and missed values were imputed by the robust sequential imputation of missing values [47]. Then, we performed differential expression analysis using the limma package [48] and pathway enrichment analyses of differentially expressed proteins using “clusterProfiler” [49], “Ggplot2” [50], “EnhancedVolcano” [51], and “pheatmap” [52] packages for visualization.

### 4.12. Y397-FAK Autophosphorylation Inhibition

FAK Inhibitor 14 (pFAKi; 1 µM; Abcam, Cambridge, UK), a selective inhibitor of Y397-FAK autophosphorylation, was used to determine the dependence of FA maturation and the following FAK autophosphorylation on CSF behavioral changes during culturing on fibrillar substrates. CSFs (1 × 10^4^ cm^−2^) were first seeded on TCPS and cultured in standard DMEM/F-12 medium for 24 h. The standard medium was replaced by DMEM/F-12 medium containing 1 µM of FAK Inhibitor 14 and changed daily for three days to pre-inhibit FAK autophosphorylation. After that, CSFs were gently passaged, seeded on prepared scaffolds, and cultured in DMEM/F-12 medium containing 1 µM of FAK Inhibitor 14. The influence of pFAKi on CSF behavior during culturing on fibrillar substrates was then assessed via the analysis of FA maturation, mechanotransduction initiation, ECM remodeling-associated protein expression, and cell motility changes, as described in the sections above.

### 4.13. Western Blot Analysis

To detect FAK and pFAK protein levels, cells were first washed several times with PBS (pH 7.4, Ca^2+^ and Mg^2+^ free) at 37 °C and lysed in 50 µL of RIPA lysis buffer (Sigma-Aldrich) containing Halt^TM^ protease and phosphatase inhibitor cocktail (Thermo Scientific) for 5 min on ice with stirring. Cell lysates were collected with the rubber scraper and centrifuged at 15,000× *g* for 10 min. After that, supernatants were diluted in a ratio of 1:1 with 2× Laemmli sample buffer containing 3% dithiothreitol and incubated at 90 °C for 5 min. The protein concentration in samples was adjusted by the dot–blot method, where direct application of the samples to a PVDF membrane (Sigma-Aldrich) was followed by staining with Coomassie Brilliant blue R-250 (Sigma-Aldrich) and the analysis of protein concentration with the Quantity One software (version 4.6.6; Bio-Rad Laboratories). Proteins in lysate samples were separated by electrophoresis in polyacrylamide gel in the presence of sodium dodecyl sulfate and transferred to PVDF membranes in tris-glycine buffer (pH 8.3) containing 10% ethanol using a wet blotting system. After that, PVDF membranes were blocked for 1 h at room temperature with 0.05% PBST containing 5% BSA and incubated overnight at 4 °C with primary antibodies (Appendix A) diluted in 0.05% PBST having 5% BSA according to the manufacturer’s recommendations. After extensive washing with PBST, PVDF membranes were incubated with peroxidase-conjugated secondary antibodies (1:5000; Appendix A) for 1 h at room temperature, followed by washing with PBST and developing with a SuperSignal^®^ West Dura Extended Duration Substrate (Thermo Scientific). Chemiluminescence was recorded using a ChemiDoc system (Bio-Rad Laboratories).

### 4.14. Cell Viability Assay

CSF viability was assessed using the resazurin assay. CSFs (2 × 10^4^ cm^−2^) were plated in 96-well plates and allowed to adhere overnight. Following pFAKi treatments, DMEM/F12 medium containing 10% resazurin reagent (0.5 mg/mL, Sigma-Aldrich) was added to each well. Plates were then incubated at 37 °C for 4 h, allowing the metabolically active cells to reduce the resazurin reagent. After incubation, the absorbance of each well was measured using a Varioskan™ LUX multimode microplate reader (Thermo Scientific) at two wavelengths: 570 nm (reduced form) and 600 nm (oxidized form). Viability percentages were then calculated by comparing the absorbance values of treated samples to those of untreated controls, which were taken as representing 100% viability.

### 4.15. Statistical Analysis

In all experiments, at least three independent measurements were performed. Error bars are represented as the standard deviation (S.D.) of the mean, analyzed a priori for homogeneity of variance. Replicates from each independent experiment were confirmed to follow a Gaussian distribution. Differences between groups were determined using an unpaired t-test and one-way or two-way analysis of variance (ANOVA) followed by Tukey’s, Dunnett’s, Bonferroni’s, or Šidák’s multiple comparison post hoc test. Significance between groups was established for *p* < 0.05 (*), *p* < 0.001 (**), *p* < 0.0002 (***), and *p* < 0.0001 (****) with a 95% confidence interval. All statistical calculations and graph plotting were performed using Prism 9.5.0 (GraphPad Software, San Diego, CA, USA).

## Figures and Tables

**Figure 1 ijms-25-08601-f001:**
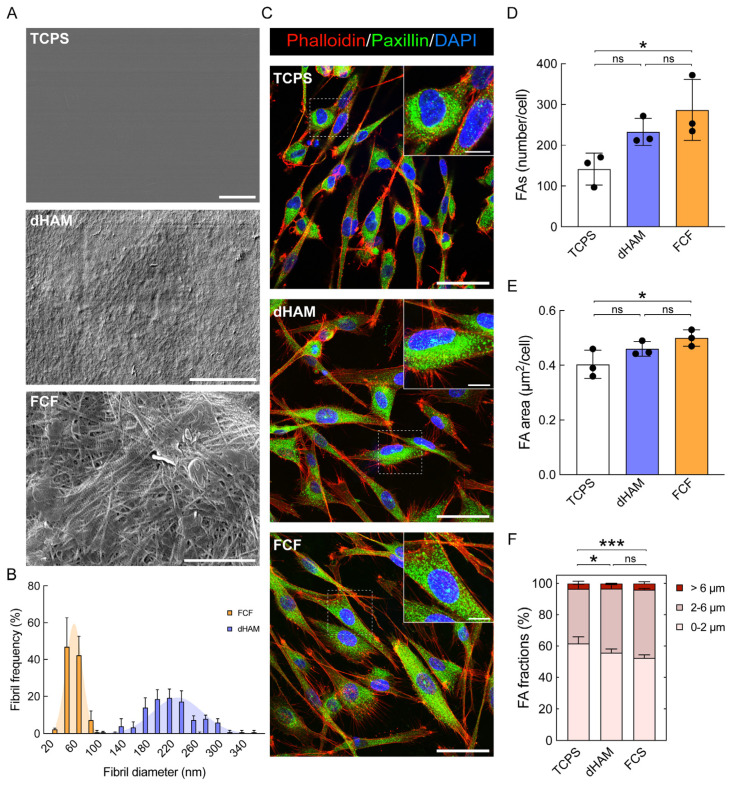
Substrate characterization and the impact of fibrillar substrates on focal adhesion (FA) maturation in human corneal stromal fibroblasts (CSFs). (**A**) Scanning electron micrographs of TCPS, dHAM, and FCF substrates. Scale bars: 2 (TCPS, FCF) and 20 (dHAM) µm. (**B**) Average frequency ± S.D. of the fibril diameter, Ø (nm), of dHAM (blue bars) and FCF substrates (orange bars), representing at least 150 measured fibrils each. The frequency histograms of dHAM and FCF substrates were used to calculate Gaussian curves through non-linear regression (blue and orange areas, respectively), with corresponding Ø = 224.2 nm ± 43.05 and 51.59 nm ± 17.61 (*n* = 3; adjusted *R*^2^ for dHAM and FCF corresponds to 0.9299 and 0.9937, respectively). (**C**) Fluoresce images of CSFs stained for actin cytoskeleton (visualized with Phalloidin-TRITC, red), paxillin (green), and DAPI (blue). Scale bars: 50 µm, 10 µm (insets). (**D**–**F**) Fibrillar substrates affect FA rearrangements in cultured CSFs. The bar graphs are represented as averages ± S.D. (*n* = 3; * corresponds to *p* < 0.05 after one-way ANOVA followed by Tukey’s multiple comparisons post hoc test). FA fractions are represented as averages ± S.D. from three independent experiments (*n* = 3; * corresponds to *p* < 0.05 and *** corresponds to *p* < 0.0002 after two-way ANOVA followed by Tukey’s multiple comparisons post hoc test). Quantitative immunofluorescence analysis (q-IFA).

**Figure 2 ijms-25-08601-f002:**
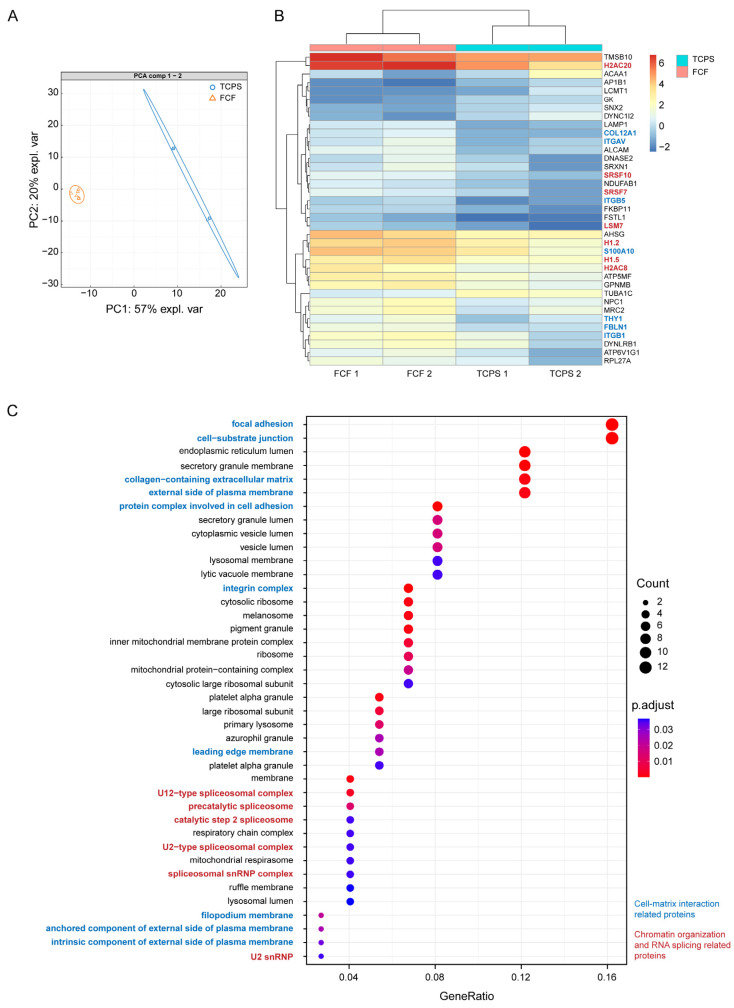
Fibrillar collagen film (FCF) affects protein expression in cultured human corneal stromal fibroblasts (CSFs). (**A**) Principal component analysis (PCA) of protein expression inherent to CSFs cultured on TCPS and FCF for 8 days (*n* = 2, three technical replicates are presented for each sample). (**B**) Heatmap of top differentially expressed proteins between CSFs cultured on TCPS and FCF. (**C**) The gene ontology (GO) cellular compartment enrichment analysis of up-regulated proteins specific for CSFs cultured on FCF.

**Figure 3 ijms-25-08601-f003:**
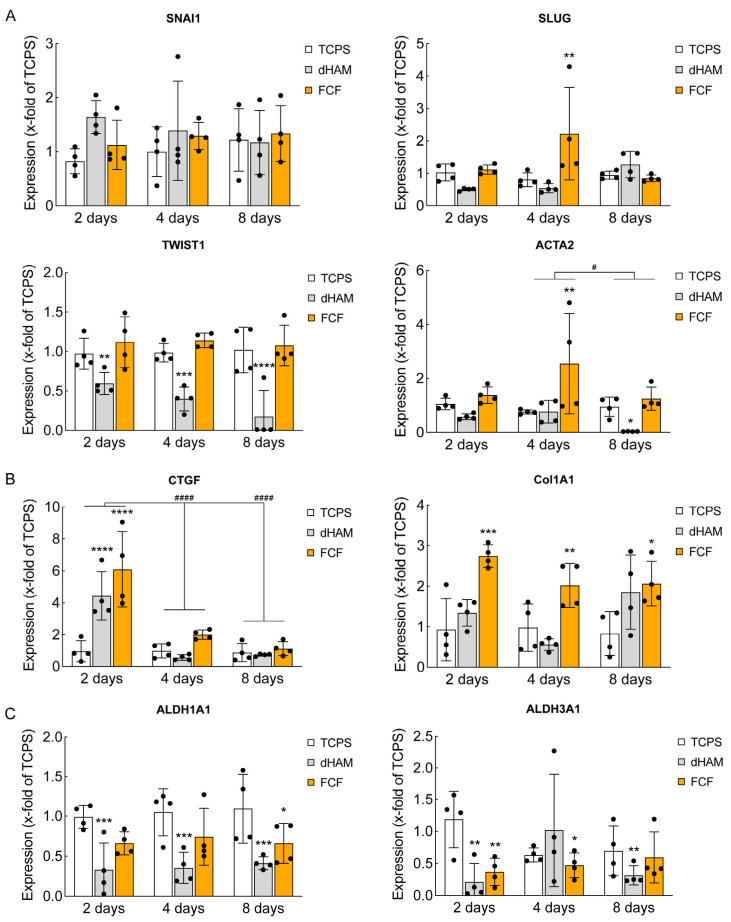
Epithelial-to-mesenchymal transition (EMT)-related and extracellular matrix (ECM) remodeling-associated gene expression features in human corneal stromal fibroblasts (CSFs) cultured on fibrillar substrates. (**A**) *SNAI1*, *SLUG*, *TWIST1*, and *ACTA2* mRNA expression during 8 days of culturing. The bar graphs are represented as averages ± S.D. and derived from four independent experiments (*n* = 4; * corresponds to *p* < 0.05, ** corresponds to *p* < 0.001, *** corresponds to *p* < 0.0002, and **** corresponds to *p* < 0.0001 after two-way ANOVA followed by Dunnett’s multiple comparisons post hoc test compared to TCPS). Additional statistical analysis was conducted to compare gene expression differences at 4 and 8 days with those at 2 days (# corresponds to *p* < 0.05 after two-way ANOVA followed by Tukey’s multiple comparisons post hoc test). (**B**) *Col1A1* and *CTGF* mRNA expression during 8 days of culturing. The bar graphs are represented as averages ± S.D. and derived from four independent experiments (*n* = 4; * corresponds to *p* < 0.05, ** corresponds to *p* < 0.001, *** corresponds to *p* < 0.0002, and **** corresponds to *p* < 0.0001 after two-way ANOVA followed by Dunnett’s multiple comparisons post hoc test compared to TCPS). Additional statistical analysis was conducted to compare gene expression differences at 4 and 8 days with those at 2 days (#### corresponds to *p* < 0.0001 after two-way ANOVA followed by Tukey’s multiple comparisons post hoc test). (**C**) *ALDH1A1* and *ALDH3A1* mRNA expression during 8 days of culturing. The bar graphs are represented as averages ± S.D. and derived from four independent experiments (*n* = 4; * corresponds to *p* < 0.05, ** corresponds to *p* < 0.001, and *** corresponds to *p* < 0.0002 after two-way ANOVA followed by Dunnett’s multiple comparisons post hoc test compared to TCPS). Quantitative real-time polymerase chain reaction (qRT-PCR). The data are normalized to *GAPDH* and calculated using the ∆∆Ct method.

**Figure 4 ijms-25-08601-f004:**
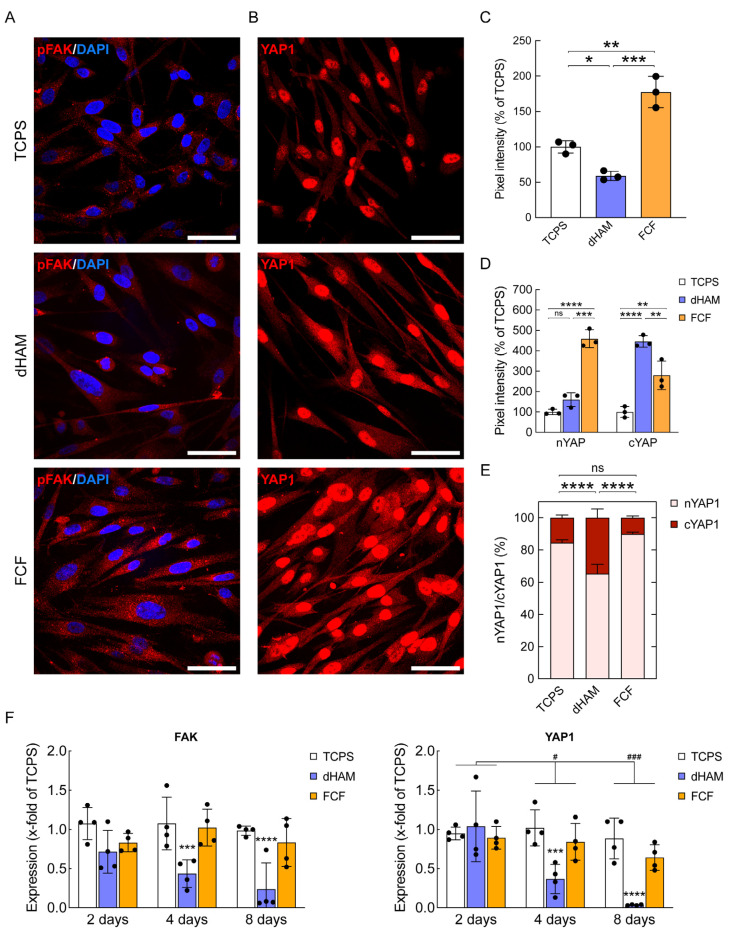
Fibrillar collagen film (FCF) initiates early mechanotransduction in human corneal stromal fibroblasts (CSFs). (**A**) Fluoresce images of CSFs stained for pFAK (red) and DAPI (blue). Scale bars: 50 µm. (**B**) Fluoresce images of CSFs stained for YAP1 (red). Scale bars: 50 µm. (**C**) Comparison of pFAK pixel intensity relative to TCPS. The bar graphs are represented as averages ± S.D. and derived from three independent experiments (*n* = 3; * corresponds to *p* < 0.05, ** corresponds to *p* < 0.001, *** corresponds to *p* < 0.0002, and **** corresponds to *p* < 0.0001 after one-way ANOVA followed by Tukey’s multiple comparisons post hoc test). q-IFA. (**D**) Fibrillar substrates result in early YAP1 nuclear translocation in cultured CSFs. The bar graphs are represented similarly to (**C**). (**E**) Nuclear YAP1 (nYAP1) to cytoplasmic YAP1 (cYAP1) ratio. The ratio is represented as an average ± S.D. from three independent experiments (*n* = 3; **** corresponds to *p* < 0.0001 after two-way ANOVA followed by Tukey’s multiple comparisons post hoc test). Quantitative immunofluorescence analysis (q-IFA). (**F**) *FAK* and *YAP1* gene expression features in CSFs cultured on fibrillar substrates for 8 days. The bar graphs are represented as averages ± S.D. and derived from four independent experiments (*n* = 4; *** corresponds to *p* < 0.0002 and **** corresponds to *p* < 0.0001 after two-way ANOVA followed by Dunnett’s multiple comparisons post hoc test compared to TCPS). Additional statistical analysis was conducted to compare gene expression differences at 4 and 8 days with those at 2 days (# corresponds to *p* < 0.05 and ### corresponds to *p* < 0.0002 after two-way ANOVA followed by Tukey’s multiple comparisons post hoc test). Quantitative real-time polymerase chain reaction (qRT-PCR). The data are normalized to *GAPDH* and calculated using the ∆∆Ct method.

**Figure 5 ijms-25-08601-f005:**
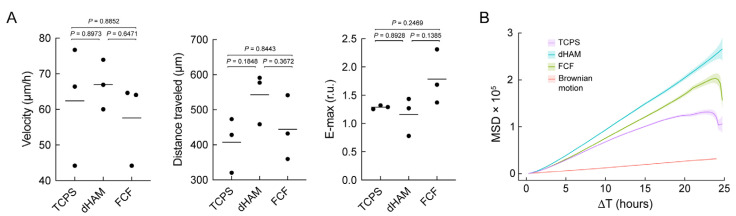
Migration pattern changes in corneal stromal fibroblasts (CSFs) cultured on fibrillar substrates. (**A**) Average values, representing cell velocity, distance traveled, and the degree of cell movement (E-max), estimated for CSFs cultured on TCPS, dHAM, and FCF during 24 h (*n* = 3 with 150 cells analyzed per each biological repeat; one-way ANOVA followed by Tukey’s multiple comparisons post hoc test). (**B**) Mean square displacement (MSD) of CSFs cultured on different substrates for 24 h.

**Figure 6 ijms-25-08601-f006:**
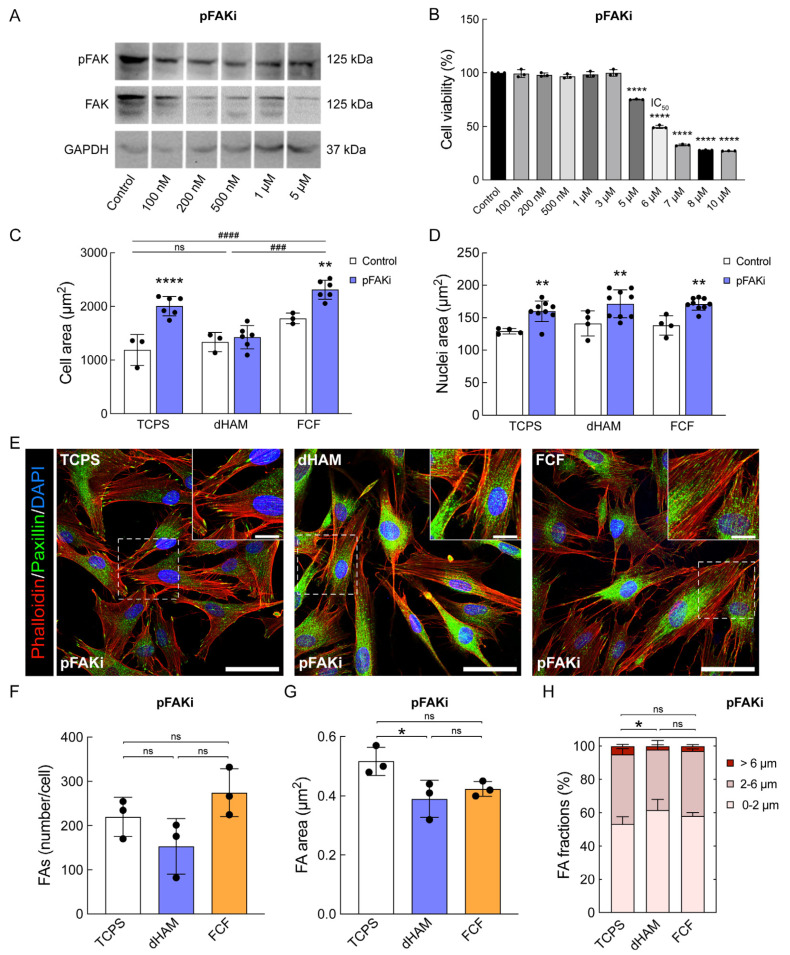
FAK Inhibitor 14 (pFAKi) induces cell spreading and affects focal adhesion (FA) maturation in cultured human corneal stromal fibroblasts (CSFs). (**A**) Effect of pFAKi on FAK and pFAK protein levels in CSFs on TCPS after 3 days of culturing. (**B**) CSFs’ viability after 3 days of culturing with various pFAKi concentrations (IC50 = 6 µM; *n* = 3; **** corresponds to *p* < 0.0001 after one-way ANOVA followed by Dunnett’s multiple comparisons post hoc test compared to control). (**C**,**D**) pFAKi results in an increase in cell and nuclei areas in cultured CSFs. The bar graphs are represented as averages ± S.D. (Cell area: *n* = 3, Control; *n* = 6, pFAKi; Nuclei area: *n* = 4, Control; *n* = 9, pFAKi; ** corresponds to *p* < 0.001 and **** corresponds to *p* < 0.0001 after two-way ANOVA followed by Bonferroni’s multiple comparisons post hoc test compared to control). Additional statistical analysis was conducted to compare different substrates (### corresponds to *p* < 0.0002 and #### corresponds to *p* < 0.0001, two-way ANOVA followed by Bonferroni’s multiple comparisons post hoc test). Quantitative immunofluorescence analysis (q-IFA). (**E**) Fluoresce images of CSFs stained for actin cytoskeleton (visualized with Phalloidin-TRITC, red), paxillin (green), and DAPI (blue). Scale bars: 50 µm, 10 µm (insets). (**F**–**H**) pFAKi affects FA maturation on fibrillar substrates. The bar graphs are represented as averages ± S.D. and derived from three independent experiments (*n* = 3; * corresponds to *p* < 0.05 after one-way ANOVA followed by Tukey’s multiple comparisons post hoc test). FA fractions are represented as averages ± S.D. from three independent experiments (*n* = 3; * corresponds to *p* < 0.05 after two-way ANOVA followed by Tukey’s multiple comparisons post hoc test). Quantitative immunofluorescence analysis (q-IFA).

**Figure 7 ijms-25-08601-f007:**
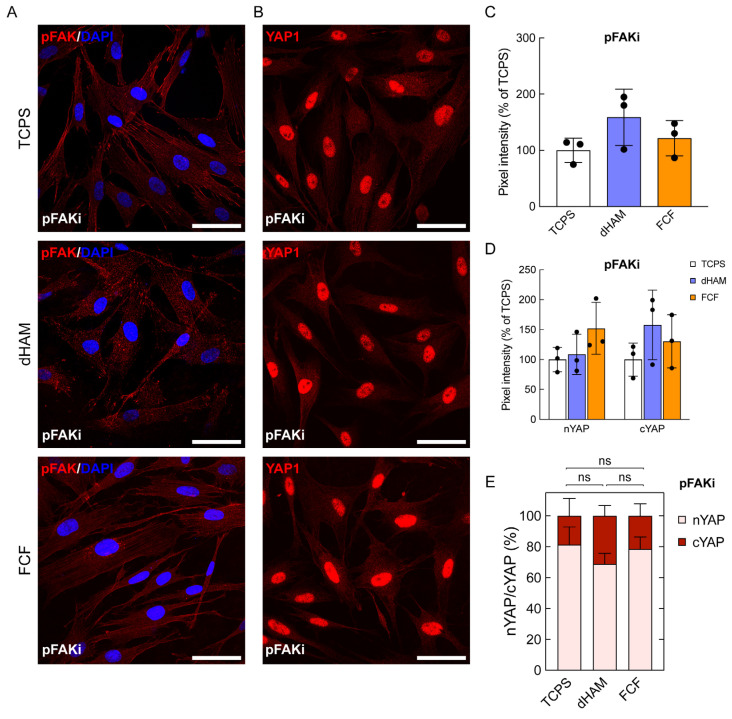
FAK inhibition (pFAKi) overrules early mechanotransduction in human corneal stromal fibroblasts (CSFs) cultured on fibrillar collagen film (FCF). (**A**) Fluoresce images of CSFs stained for pFAK (red) and DAPI (blue). Scale bars: 50 µm. (**B**) Fluoresce images of CSFs stained for YAP1 (red). Scale bars: 50 µm. (**C**) pFAKi suppresses FAK autophosphorylation during the first 8 h. The bar graphs are represented as averages ± S.D. and derived from three independent experiments (*n* = 3; *p* = 0.2150, one-way ANOVA followed by Tukey’s multiple comparisons post hoc test). Quantitative immunofluorescence analysis (q-IFA). (**D**,**E**) pFAKi prevents YAP1 nuclear translocation in cultured CSFs. The bar graphs are represented as averages ± S.D. and derived from three independent experiments (*n* = 3; *p* = 0.2047, two-way ANOVA followed by Tukey’s multiple comparisons post hoc test). The nuclear YAP1 (nYAP1) to cytoplasmic YAP1 (cYAP1) ratio is represented as an average ± S.D. from three independent experiments (*n* = 3; two-way ANOVA followed by Tukey’s multiple comparisons post hoc test). Quantitative immunofluorescence analysis (q-IFA).

## Data Availability

The mass spectrometry proteomics data were deposited to the ProteomeXchange Consortium via the PRIDE partner repository with the dataset identifiers PXD048096 and 10.6019/PXD048096.

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
