# Peer review of "Focal Adhesion Maturation Responsible for Behavioral Changes in Human Corneal Stromal Fibroblasts on Fibrillar Substrates"

_ijms, 2024, doi:10.3390/ijms25168601_

Round 1

Reviewer 1 Report

Comments and Suggestions for Authors

This study compared how decellularized human amniotic membrane (dHAM) and bespoke fibrillar collagen scaffold (FCS) affect KC fate and behavior via the analysis of focal adhesion (FA) maturation, mechanotransduction initiation, and the following alterations in the ECM remodeling-related gene expression, protein synthesis, and behavioral changes in cultured human corneal keratocytes. It is a well-written study. However there are some questions should be clarified more.

1.          What is “GO” cellular compartment enrichment analysis…in line 115 & 120?

2.          Apart from CTGF, SNAI1 and TWIST1, do you evaluate more ECM remodeling-related or connective tissue remodeling-associated gene expression, such as fibronectin, alpha-SMA….

Author Response

We would like to thank the editor and the reviewers for their careful work reviewing the manuscript, and for the opportunity to submit a revised version.  We have gone through the specific comments from each reviewer and address each individually below (our replies are in italics). We have made changes to the manuscript accordingly, as visible in the highlighted version of the manuscript and as indicated in our replies. In addition, we have carefully read the manuscript to improve flow and readability. Overall, we feel that the manuscript has been significantly improved after this revision, and we hope that the reviewers agree.

  1. What is “GO” cellular compartment enrichment analysis…in line 115 & 120?

Thank you for indicating this oversight. We have specified the abbreviation used (p. 5, lines 123-129). GO stands for gene ontology (GO) cellular compartment enrichment analysis.

  1. Apart from CTGF, SNAI1 and TWIST1, do you evaluate more ECM remodeling-related or connective tissue remodeling-associated gene expression, such as fibronectin, alpha-SMA….

We believe that the provided gene expression analysis regarding the epithelial-to-mesenchymal transition (EMT) in keratocytes (KCs) covers an adequate number of the associated targets, which include the major genes, such as SNAI1, SLUG, TWIST1, and ACTA2 (α-SMA). At the same time, we understand that the provided range for the ECM/connective tissue remodeling-related genes is limited to CTGF and Col1A1, and additional gene expression analysis could enrich the results.

Reviewer 2 Report

Comments and Suggestions for Authors

The manuscript by Zhurenkov et al. provides a comprehensive study of the role of focal adhesion maturation in influencing the behavior of human corneal keratocytes on fibrillar substrates. The study reveals that FCS substrates can increase the initial development of FAs, which in turn triggers the activation and subsequent transmission of mechanotransduction signals. The authors are commended for an interesting study. Specific comments/suggestions to further improve the manuscript prior to acceptance are as follows:

1.     How do the stiffness levels of the dHAM and FCS substrates compare to that of native corneal tissue?

2.     Could the authors provide a more comprehensive description of the YAP analysis?

3.     Regarding Figures 1F, 4E, 6H, and 7E, could the authors conduct a statistical analysis and incorporate the results into these panels?

4.     The authors have investigated the gene expression in KCs influenced by the FCS substrates during long-term culture (spanning days). However, how does this gene expression change during short-term culture, such as within a few hours? Additionally, have the authors verified FAs to ensure that the impact of different substrates on FAs was maintained at d2, 4, and 8 of cell culture?

5.     In Figure 3, it seems that the expression of some genes varies with the progression of culture time. Could the authors conduct a statistical analysis to confirm this observation?

6.     Please revise the caption of Figure 4F.

7.     The authors have demonstrated that pFAKi enhances the cell area. Could they also provide information on its impact on cell volume?

8.     To prevent any potential confusion, could the authors include pFAKi in the labels for Figures 6 and 7?

Author Response

We would like to thank the editor and the reviewers for their careful work reviewing the manuscript, and for the opportunity to submit a revised version.  We have gone through the specific comments from each reviewer and address each individually below (our replies are in italics). We have made changes to the manuscript accordingly, as visible in the highlighted version of the manuscript and as indicated in our replies. In addition, we have carefully read the manuscript to improve flow and readability. Overall, we feel that the manuscript has been significantly improved after this revision, and we hope that the reviewers agree.

  1. How do the stiffness levels of the dHAM and FCS substrates compare to that of native corneal tissue?

We have indicated the Young’s modulus parameters for the investigated materials (p. 2, lines 77-80) and added characterization to the supplementary information (Figure S5). The Young’s moduli accounted for 51.05 ± 17.48 and 39.51 ± 15.69 kPa for dHAM and FCS, respectively. These values closely resemble the mechanical properties of the human anterior corneal stroma, which was indicated to range from 24 to 39 kPa by the AFM.1

  1. Last, J.A.; Thomasy, S.M.; Croasdale, C.R.; Russell, P.; Murphy, C.J. Compliance Profile of the Human Cornea as Measured by Atomic Force Microscopy. Micron 2012, 43, 1293–1298, doi:10.1016/j.micron.2012.02.014.

  1. Could the authors provide a more comprehensive description of the YAP analysis?

Thank you for pointing out this oversight. In the YAP1 analysis, we subtracted the fluorescence signals of the cytoplasm and nuclei from one another. Subsequently, we measured the pixel intensities for these cellular compartments. We have specified regarding the YAP1 analysis (p. 16, lines 523-526) and added the procedure to the supplementary information (Figure S3).

  1. Regarding Figures 1F, 4E, 6H, and 7E, could the authors conduct a statistical analysis and incorporate the results into these panels?

Thank you for the suggestion. We have added statistical analysis to all mentioned figures.

  1. The authors have investigated the gene expression in KCs influenced by the FCS substrates during long-term culture (spanning days). However, how does this gene expression change during short-term culture, such as within a few hours? Additionally, have the authors verified FAs to ensure that the impact of different substrates on FAs was maintained at d2, 4, and 8 of cell culture?

We have not conducted gene expression analysis within the first hours of culturing. At the same time, we acknowledge that changes in FAK and YAP1 gene expression could be observed at the earlier stages (which we indicated in the Discussion section, p. 13, lines 360-364), as we did not see their up- or down-regulation after 2 days of culturing. Other genes analyzed did not see pronounced alterations after 2 days of culturing compared to tissue culture polystyrene (TCPS), with only CTGF and Col1A1 having substantial up-regulation.

Regarding the FA, we additionally collected data for KCs after 24 hours of culturing, which we have indicated in the manuscript (see p. 4, lines 110-112 for detail) and added to the supplementary information (Figure S6). This data continues, indicating the positive impact of FCS on cultured KCs. At the same time, we indicated that there was a substantial increase in the number of FAs in KCs cultured on TCPS. Additionally, our proteome results indicated FAs as one of the most involved cellular compartments in KCs cultured on FCS after 8 days of culturing compared to TCPS. For instance, vinculin (VCL) and talin (TLN1) saw an up-regulation of 2.4- and 2-fold, respectively, in KCs after 8 days of culturing on FCS (see supplementary proteome data for detail).

  1. In Figure 3, it seems that the expression of some genes varies with the progression of culture time. Could the authors conduct a statistical analysis to confirm this observation?

We added the corresponding statistical analysis initially. We also tried to improve the figure captions for better readability (see p. 6-7, lines 150-152 and lines 155-157). It is indicated with # symbols and represents changes on 4 and 8 days compared to 2 day results. In other comparisons, statistical significance between different days was not identified.

  1. Please revise the caption of Figure 4F.

Thank you for pointing out this oversight. We have revised the corresponding figure caption for better readability.

  1. The authors have demonstrated that pFAKi enhances the cell area. Could they also provide information on its impact on cell volume?

Thank you for your suggestion. We acknowledge that cell area and volume results can differ significantly. To address the uneven surfaces of the materials used (dHAM and FCS), we collected images using z-sections (3 × 2.5 μm) to capture most of the cells. However, it is not possible to accurately calculate cell volume based on these parameters, as the z-stack consists of a small number of large steps.

  1. To prevent any potential confusion, could the authors include pFAKi in the labels for Figures 6 and 7?

Thank you for the suggestion. We have updated the labeling to the mentioned figures.

Round 2

Reviewer 2 Report

Comments and Suggestions for Authors

The revised manuscript looks good!

Author Response

We would like to thank the academic editor and the reviewers for their careful work reviewing the manuscript. We have gone through the specific comments from the academic editor and address each individually below (our replies are in italics). We have made changes to the manuscript accordingly, as visible in the highlighted version of the manuscript and as indicated in our replies. In addition, we have carefully read the manuscript to improve flow and readability. Overall, we feel that the manuscript has been significantly improved after the revisions.

Academic Editor.

  1. The used cells are not keratocytes. They are termed corneal stromal fibroblasts because the culture is performed in the presence of serum. In these conditions, the keratocytes undergo the first activation process where they change their morphology from stellate to elongated and decrease production of marker proteoglycans. The authors need to change "keratocytes" to "fibroblasts" throughout the text and in the title.

Thank you for the suggestion. We have changed the terminology wherever it was used in the text, SI, and title.

  1. Please choose another abbreviation for fibrillar collagen scaffold. FCS is routinely used in the literature for fetal calf serum. Another unusual term that the authors might like to avoid is "bespoke" that comes from the clothing industry.

Thank you for pointing this out. We have changed the term for fibrillar collagen to fibrillar collagen film (FCF). We hope this abbreviation will better differentiate this material from potential mix-ups. We also removed the word ‘bespoke’ wherever it was mentioned.

  1. Please also specify in the Methods whether PBS used for tissue and cell washes contained calcium and magnesium.

Thank you for the recommendation. We indicated that PBS used was Ca2+ and Mg2+ free in the methodology (p. 14, lines 441-442; p. 15, line 462 and 475; p. 16, line 504; and p. 17, line 562; p. 18, line 653).

Additionally, we improved the Figure 6C and 6D caption for better readability.